# Willingness to pay for diabetic retinopathy screening in Qujiang District, rural Guangdong, southern China: a cross-sectional study

Baixiang Xiao [1,2] Xuejun Gu,[1] Ling Jin,[3] Ving Fai Chan [2,4] Yanping Li,[1] Carlos Price-Sanchez,[2] Yuanping Liu,[3] Yanfang Wang,[3] Haoxiang Fu,[5] Dongfeng Li,[2] Nathan Congdon[2,3,6]

[1]Affiliated Eye Hospital of Nanchang University, Nanchang City, China
[2]Centre for Publich Health, Queen's University Belfast, Belfast, UK
[3]The State Key Laboratory of Ophthalmology, Zhongshan Ophthalmic Center, Sun Yat-sen University, Guangzhou, China
[4]College of Health Sciences, University of KwaZulu Natal, Durban, South Africa
[5]Zhenjiang District Hospital, Shaoguan, Guangdong Province, China
[6]Orbis International, NY, New York, USA

**Correspondence to**
Dr Nathan Congdon;
ncongdon1@gmail.com

## ABSTRACT

**Objective** To determine willingness to pay for a diabetic retinopathy screening, and its determinants, among people with diabetes mellitus in Qujiang District of Shaoguan City, rural Guangdong, southern China.

**Design** This cross-sectional study was conducted through a large-scale screening programme in 2019. We randomly selected 575 (21.5%) among 2677 people over 18 years old with known diabetes who attended the screening. Participants elected to pay nothing or RMB10–RMB120 (US$1.6–US$18.8), in RMB10 intervals, displayed on printed cards. One trained interviewer collected all the data.

**Setting** Ten primary health centres in Qujiang District of Shaoguan City, Guangdong.

**Participants** 545 from the 575 randomly selected people (94.8%) agreed to participate in the study.

**Outcome measures** Proportion of participants willing to pay anything for screening, mean amount they were willing to pay and determinants of these figures.

**Results** Among 545 participants (mean age 64.6 years (SD±10.4), 40.7% men), 327 (60.0%) were willing to pay something for screening, of whom 273 (83.5%) would pay RMB10–RMB30 (US$1.6–US$4.7). People living in rural areas and those from lower-income families were more likely to be willing to pay anything, while men, urban residents and those covered by employer-linked insurance were willing to pay larger sums (p<0.05 for all).

**Conclusion** Nearly two-thirds of participants were willing to pay for screening in this screening programme organised at the primary care level in rural China. This finding offers the potential that such activities can be sustained and scaled up through user fees.

## STRENGTHS AND LIMITATIONS OF THIS STUDY

⇒ The study was nested in a large-scale, real-world screening programme in rural China, an area where little is known about willingness to pay for diabetic eye screening services.
⇒ The use of payment cards of differing amounts, presented simultaneously to participants for selection, avoided starting point bias in bidding for payment.
⇒ The upper and lower limits of amounts the study offered for participants' selection were based on the actual cost of retinal detection in hospital and in the community.
⇒ The main weakness is that study was conducted at a single location and can only be applied to other settings with caution.

## INTRODUCTION

The global number of people living with diabetes mellitus (PwDM) has increased steadily from 108 million in 1980 to 425 million in 2017.[1] It is expected to increase to 693 million cases by 2045.[2] The increasing burden of PwDM in China has followed international trends, with the prevalence of adult PwDM increasing from approximately 2% in the 1980s to over 10% in 2017.[3–5] This dramatic rise in PwDM is due to increases in life expectancy, wider adoption of western diets, sedentary lifestyles and expanding urbanisation.[5]

While diabetes mellitus (DM) itself promises to grow as a health concern, complications associated with its progression pose additional risks to health. DM can lead to ocular complications requiring further ophthalmological care, straining the global public health system. Of these ocular conditions, Diabetic retinopathy (DR) is the most common and serious, potentially resulting in irreversible blindness (BL). After 20 years, 90% of persons with type 1 diabetes and 60% with type 2 will develop some form of retinopathy. One-third of these cases will require treatment to prevent irreversible BL.[6] Given current numbers of PwDM and the prevalence of DR, the need for laser treatments is estimated to be over 8 million in China at present.[5 7]

Interventions at the level of primary care are effective at both reducing the severity and

slowing the progression of DR. Methods such as adopting a healthy lifestyle to control blood glucose, adhering to prescribed medicines, undergoing regular retinal examinations and receiving timely treatment for retinopathy with laser photocoagulation, intravitreal injections or vitrectomy for more serious cases, have together been proven to reduce BL from DR by over 90%.[8 9] In particular, establishing robust DR screening (DRS) protocols at the level of primary care is a key step in preventing the progression of DR towards visual impairment (VI) or BL.

Despite the efficacy of DRS in preventing vision impairment, increasing the uptake of DRS has been a significant challenge for many health systems.[10 11] Common barriers to uptake include: a lack of perceived need, patient fear of positive diagnosis, high cost, lack of access and low quality of service.[10] In addition, DR is often asymptomatic before the disease affects the macula or causes a vitreous haemorrhage and begins to impair visual acuity (VA). Therefore, many unscreened patients are unaware they have the condition before it becomes severe.[12–17] As DR becomes more prevalent, integrating DRS at all levels of service has never been more important, but its benefits may not be obvious to asymptomatic patients.

As part of a pilot programme to develop and enhance primary and secondary healthcare models for DRS, Orbis International and the World Diabetes Foundation have supported eye care services for PwDM at six county hospitals in Guangdong Province since 2017. The current willingness to pay (WTP) study was conducted as part of the DRS programme in Qujiang District of Shaoguan City, northern Guangdong Province, China in the second half of 2019. Though prior studies in the area have examined WTP for cataract surgery,[18] at present, no such studies on DRS have been completed in China. This study aims to fill this gap by exploring WTP for DRS services in rural North Guangdong Province. Despite the current lack of targeted funding for DRS in most areas of China, charging for this crucial service might be sufficient to sustain the service if beneficiaries are WTP enough. The findings in this report will provide context for future decisions regarding providing care for PwDM and establishing community DRS services in rural China.

## METHODS

### Study design

This cross-sectional study was conducted through a large-scale screening programme in 2019. PwDM found with sight-threatening DR (R2 or above in the UKNHS DRS system),[19] operable cataracts, or other suspected eye diseases were referred to secondary-level hospitals for further examination or treatment. The study fulfilled the tenets of the Declaration of Helsinki.

### Setting and participants

Qujiang is a secondary-level district in North Guangdong province of China, home to 320 000 people in 2018 (a population density of 192 people per km,[2] with a 2018

per capita gross domestic product (GDP) of US$7160.32. Qujiang's GDP is significantly less than the national average per capita GDP of US$10 253.97 in the same year. These figures are typical for other poor rural areas in southern China.

Shaoguan City Railway Hospital and Qujiang County Hospital are secondary-level hospitals in Zhenjiang and Qujiang Districts, respectively. Two doctors from the eye department of Shaoguan City Railway hospital were trained in detecting and treating DR with fundus laser and intravitreous injection. Trainers for DRS and managers were also trained in the programme to arrange periodic, small-scale DRS in rural township health facilities and urban community health centres since 2017. Basic examinations for anterior segment eye disease were provided at the outpatient department in Qujiang Hospital. Patients suspected of having posterior segment diseases, including DR, would have to travel to the neighbouring district hospital in Zhenjiang or a facility in the city centre.

As in many other places in China, PwDM are encouraged to register with the primary health system in Qujiang District for a free annual health check, monitoring of complications and health education. Village doctors (VDs) provide free fasting blood glucose checks for people aged over 35 years or anyone wishing to test their glucose levels. People suspected of having DM are referred for further examination at secondary-level hospitals. A total of 3646 PwDM registered across 10 primary health centres in Qujiang District were informed of these local health bureau-supported DRS services, at which point they were able to schedule appointments with trained VDs.

Study team members at the screening sites included an ophthalmologist, two nurses, one technician and one interviewer, all trained for the study. PwDM came to the township health units for the screening at a prescheduled time.

Of the 3646 PwDM informed about the screening, 2677 (73.4%) from across the 10 towns in Qujiang attended. This was the first formally arranged, large-scale DRS conducted in the district. We randomly approached 575 PwDM (21.5%) attending the DRS by generating a random number for the selection of a first participant and then approaching every sixth participant from that point on who arrived for screening. In total, 545 (94.8%) of those approached agreed to participate in the WTP study and were recruited.

### Data collection

All participants were informed of the scheduled DRS 1–2 months before screening started, through the trained VDs in the study area and posters circulated in the villages. They were informed of the WTP study by the interviewer at the screening site. Draft questionnaires were tested with some PwDM.

Participants were welcomed to the scheduled annual DR screening by their local public health doctor at each

screening site. The local physician would then personally inform each participant when their screening reports were available via mobile phone, text or WeChat message within 1 week. Participants were then supplied with DR educational bulletins and posters, which provided information on DR and highlighted the importance of receiving annual fundus imaging for the early detection of ocular diseases.

Participants' basic information and disease history were collected and double-checked via the Chronic Disease Recording System. Retrieved data included participant name, age, sex, site of residence (rural or urban), education, profession, insurance, self-assessment of VA, economic situation including the proportion of family income spent on food, and history of diseases including DM and hypertension. Venous blood was collected from each participant to monitor Glycated hemoglobin (HbA1c). Presenting VA was tested monocularly in each eye at 6 m with a retro-illuminated Snellen visual chart before the ophthalmologist carried out a slit lamp examination. Intraocular pressure (IOP) was measured in each eye using tono-pen before pupils were dilated with compound tropicamide to enable the capture of fundus images with a Canon CR-2 by a trained nurse. Patients with high IOP or shallow anterior chambers detected under slit lamp examination were not dilated; instead, images were taken through pupils which had been dilated scotopically after around 3–5 min in a darkened room.

Diabetic complications were defined as any of the following symptoms: ulcers on hands or feet, cardiovascular disease, diabetic nephropathy, diabetic eye disease, ketoacidosis or other serious metabolic complications. Participants could self-assess their physical health and VA on a scale of 1–5, corresponding to: bad (1), not good (2), alright (3), good (4) and excellent (5). Regarding their economic situation, respondents were given two choices related to the proportion of monthly family income spent on food: less than 30%, equal to or over 30%.

Questionnaires regarding WTP for the DR screening were administered post screening in a quiet, separate room. Each participant was given the opportunity to ask any questions they might have on DM, DR, screening or the report, and received informed responses from the trained interviewer. Comments on the screening were also invited. The participant was then told that the screening required appropriate human resources, equipment and preparation, so there was some cost. The direct cost (travel or transport cost, daily allowance for screening staff) of one person screened was approximately US$4.7. At the time of the study, there was no budget in the primary healthcare system for the DRS. According to the provincial pricing regulations, fundus imaging charges were US$18.8 for both eyes at secondary level hospitals. Ultimately, these costs fall on the patient, as expenses associated with imaging and registration are not currently reimbursed by insurance.

Each participant was informed that their choice of WTP category would not affect their ability to receive primary healthcare, medical services in hospitals, nor current or any future DR screening. Participants were first asked whether they would like to pay something, starting at ¥10 (US$1.6), for the screening. Reasons were elicited for those answering 'no'. When a participant expressed WTP any amount, they were presented with cards marked from ¥10 (US$1.6) to ¥120 (US$18.8) at intervals of ¥10 (US$1.6) (12 cards in total). To avoid bargaining bias, we presented participants with cards valued at RMB10, RMB20, RMB30, RMB40, RMB50, RMB60, RMB70, RMB80, RMB90, RMB100, RMB110 and RMB120 simultaneously for their selection. A trained interviewer collected all data for WTP to maintain the consistency and quality of the interview, although general data were collected by three trained staff.

## Statistical analysis

Participants' characteristics were presented as mean (SD) for continuous data with normal distribution or median (IQR) for data without normal distribution, and frequency (%) for categorical variables. Shapiro-Wilk normality tests and histograms were applied to check the normality of continuous data. Participants' age was categorised by intervals of 10 years. Duration of diabetes was categorised by intervals of 5 years. The cut-offs of 6/60, 6/18 and 6/12 were used to describe VA in the better-seeing eye.

Logistic regression analysis for WTP and linear regression analysis for the different amounts participants would pay were employed to examine the associations between potential factors and outcomes. Univariable and multivariable regression models were fitted. Age, sex and variables with $p < 0.20$ in the univariable analyses were included in the multivariable regression model. All statistical analyses were performed using a commercially available software package (Stata V.16, StataCorp).

The reasons given by the participants who would not pay anything for the DRS at community level were listed in text including all the points mentioned.

## Patient and public involvement

Participants were not directly involved in designing or implementing the study, although draft questionnaires were tested with some PwDM.

## RESULTS

Among 545 participants, the mean age was 64.4±10.4 years (range 35–92 years), and 323 (59.3%) were women (table 1). A total of 345 (63.3%) participants lived in urban settings. Almost half (44.6%) of participants had received a primary school education and 54 (9.9%) attended high school or above. All participants had either standard residential medical insurance (n=372, 68.3%) or employment-linked insurance (n=173, 31.7%).

Approximately half (51.2%) of participants had been diagnosed with DM within the last 5 years and only 63 (11.6%) had been diagnosed over 15 years

**Table 1** Demographic characteristics

| Characteristic | Mean (SD)/n (%) |
|---|---|
| Total | 545 |
| Age (year), mean (SD) | 64.6 (10.4) |
| <50 | 48 (8.8) |
| 50–59 | 131 (24.0) |
| 60–69 | 198 (36.3) |
| 70–79 | 137 (25.1) |
| 80+ | 31 (5.7) |
| Gender, n (%) | |
| Male | 222 (40.7) |
| Female | 323 (59.3) |
| Duration of DM (year), n (%) | |
| <5 | 279 (51.2) |
| 5–10 | 130 (23.9) |
| 10–15 | 73 (13.4) |
| >15 | 63 (11.6) |
| Diabetic complications present, n (%) | |
| Yes | 434 (79.6) |
| No | 111 (20.4) |
| Residence, n (%) | |
| Urban | 345 (63.3) |
| Rural | 200 (36.7) |
| Education, n (%) | |
| Never received formal education | 81 (14.9) |
| Primary school | 243 (44.6) |
| Junior high school | 167 (30.6) |
| High school or above | 54 (9.9) |
| Presenting visual acuity in better-seeing eye, n (%) | |
| ≥6/12 | 435 (79.8) |
| ≥6/18 to <6/12 | 72 (13.2) |
| ≥6/60 to <6/18 | 32 (5.9) |
| <6/60 | 6 (1.1) |
| Medical insurance, n (%) | |
| Residential medical insurance | 372 (68.3) |
| Employment-linked insurance | 173 (31.7) |
| Self-assessment of current health situation, n (%) | |
| Bad | 11 (2.02) |
| Not good | 107 (19.6) |
| Good | 261 (47.9) |
| Very good | 153 (28.1) |
| Excellent | 13 (2.39) |
| Self-assessment of current visual acuity, n (%) | |
| Bad | 12 (2.20) |
| Not good | 101 (18.5) |

Continued

**Table 1** Continued

| Characteristic | Mean (SD)/n (%) |
|---|---|
| Good | 349 (64.0) |
| Very good | 74 (13.6) |
| Excellent | 9 (1.65) |
| Economic situation, n (%) | |
| <30% of monthly family income spent on food | 108 (19.8) |
| ≥30% of monthly family income spent on food | 437 (80.2) |

DM, diabetes mellitus.

prior (table 1). Most participants (n=435, 79.8%) had presenting VA≥6/12 in the better-seeing eye. Most participants (n=434, 79.6%) did not report any diabetic complications. Nearly 80% of participants reported being in good, very good or excellent health at the time of the screening, and a similar proportion described their vision as falling in the same three categories. Regarding their economic situation, most participants (80.2%) reported their households spent 30% or above of their monthly family income on food, while 19.8% spent less than 30%. (table 1)

Overall, 327 (60.0%) participants expressed a WTP something for the DRS at the primary level, of whom 273 (83.5%) would pay RMB10–RMB30 (US$1.6–US$4.7), 34 (10.4%) would pay RMB40–RMB50 (US$6.3–US$7.8) and 15 (4.6%) would pay RMB80–RMB120 (US$12.5–US$18.8). In logistic regression models, only rural residence and those paying ≥30% of income for food (both more willing) were associated with WTP something for DRS (table 2). Both variables were significant in the univariate and multivariate models.

Among 327 PwDM who were willing to pay something for screening, in univariate linear regression models, men, urban residents and those with higher educational levels and covered by employment-linked insurance were willing to pay more (table 3). Conversely, participants with worse self-rated health and vision were willing to pay less. Only the associations with male sex, urban residence and employment-linked health insurance remained significant in the multivariate models. The level of HbA1C was not found having significant association with WTP for DRS.

Reasons given by participants unwilling to pay anything for the DRS included: should be covered by government or medical insurance; screening report was not received so cannot tell the quality of this service; suspicious whether the money charged would be used properly; suspicious whether the programme would be sustainable; would rather go to hospital.

## DISCUSSION

In this study, among 545 randomly selected people with diagnosed diabetes who attended a large scale, primary

**Table 2** Logistic regression analysis of willingness to pay for diabetic retinopathy screening, Qujiang, China (n=545)

| Characteristic | Univariable regression | | Multivariable regression* (n=545) | |
|---|---|---|---|---|
| | OR (95% CI) | P value | OR (95% CI) | P value |
| Age (per year) | 0.99 (0.97 to 1.01) | 0.217 | 1.00 (0.98 to 1.02) | 0.690 |
| Female sex | 0.83 (0.59 to 1.18) | 0.302 | 0.98 (0.66 to 1.44) | 0.903 |
| Duration of DM (year) | | | | |
| <5 | Reference | | Reference | |
| 5–10 | 0.67 (0.44 to 1.03) | 0.066 | 0.71 (0.46 to 1.11) | 0.138 |
| 10–15 | 0.64 (0.38 to 1.08) | 0.094 | 0.75 (0.43 to 1.30) | 0.306 |
| >15 | 0.85 (0.48 to 1.49) | 0.568 | 1.18 (0.64 to 2.19) | 0.591 |
| Diabetic complications | 0.89 (0.58 to 1.37) | 0.602 | | |
| Urban residence | 0.46 (0.31 to 0.66) | <0.001 | 0.44 (0.28 to 0.69) | <0.001 |
| Education | | | | |
| Never received formal education | Reference | | Reference | |
| Primary school | 1.07 (0.64 to 1.78) | 0.795 | 1.31 (0.75 to 2.28 | 0.350 |
| Junior high school | 1.16 (0.68 to 1.99) | 0.579 | 1.39 (0.74 to 2.63 | 0.310 |
| High school or above | 1.81 (0.87 to 3.75) | 0.113 | 2.07 (0.90 to 4.78 | 0.087 |
| Presenting visual acuity in better-seeing eye | | | | |
| ≥6/12 | Reference | | | |
| ≥/18 to <6/12 | 0.93 (0.51 to 1.71) | 0.823 | | |
| ≥6/60 to <6/18 | 1.03 (0.58 to 1.82) | 0.918 | | |
| <6/60 | 1.33 (0.24 to 7.36) | 0.741 | | |
| Standard residential health insurance | 1.41 (0.98 to 2.03) | 0.066 | 0.94 (0.59 to 1.51) | 0.804 |
| Self-assessment of health | | | | |
| Bad | Reference | | | |
| Not good | 0.96 (0.26 to 3.47) | 0.947 | | |
| Good | 0.70 (0.20 to 2.46) | 0.582 | | |
| Very good | 1.02 (0.29 to 3.63) | 0.978 | | |
| Excellent | 3.14 (0.45 to 22.0) | 0.248 | | |
| Self-assessment of visual acuity | | | | |
| Bad | Reference | | Reference | |
| Not good | 1.00 (0.30 to 3.38) | 0.996 | 1.09 (0.30 to 3.92) | 0.894 |
| Good | 0.96 (0.30 to 3.08) | 0.944 | 0.74 (0.21 to 2.55) | 0.629 |
| Very good | 1.80 (0.51 to 6.32) | 0.357 | 1.13 (0.29 to 4.37) | 0.854 |
| Excellent | 5.71 (0.53 to 61.4) | 0.150 | 4.03 (0.34 to 47.5) | 0.268 |
| Economic situation | | | | |
| <30% monthly income on food | Reference | | Reference | |
| ≥30% on food | 1.66 (1.09 to 2.54) | 0.018 | 1.70 (1.04 to 2.77) | 0.035 |

*Age, sex and variables with p<0.20 in the univariable regression analysis were included in the multivariable regression model.
DM, diabetes mellitus.

care DRS in rural southern China, nearly two-thirds (60%) were willing to pay something for DRS. The groups most likely to be willing to pay something were rural-dwellers and economically less-advantaged families. Among participants willing to pay something, those willing to pay more included men, urban-dwellers and those covered by employment-linked insurance. This study has highlighted a considerable WTP for these services among primary care patients.

The other WTP study for eye health services we could identify in China, conducted in 2006 on cataract surgeries, established that 73% of cataract patients were willing to pay something, and that many were willing to pay more for experienced surgeons and higher-quality intraocular lenses.[20] Possible reasons for a lower proportion willing to pay anything for DRS as compared with cataract might include the belief that DRS is a public health service and therefore should be

**Table 3** Linear regression analysis of participant willing to pay something for diabetic retinopathy screening, Qujiang, China (N=327)*

| Characteristic | Univariable regression | | Multivariable regression† (n=327) | |
| --- | --- | --- | --- | --- |
| | β (95% CI) | P value | β (95% CI) | P value |
| Age (per year) | −0.0003 (−0.003 to 0.002) | 0.847 | −0.002 (−0.004 to 0.001) | 0.281 |
| Female sex | −0.10 (−0.16 to to 0.05) | <0.001 | −0.07 (−0.13 to to 0.02) | 0.013 |
| Duration of DM (year) | | | | |
| <5 | Reference | | | |
| 5–10 | 0.02 (−0.05 to 0.09) | 0.628 | | |
| 10–15 | 0.006 (−0.08 to 0.10) | 0.890 | | |
| >15 | 0.05 (−0.04 to 0.14) | 0.283 | | |
| Diabetic complications present | 0.04 (−0.03 to 0.11) | 0.300 | | |
| Urban residence | 0.16 (0.10 to 0.21) | <0.001 | 0.14 (0.07 to 0.20) | <0.001 |
| Education | | | | |
| Never received formal education | Reference | | Reference | |
| Primary school | 0.15 (0.06 to 0.23) | 0.001 | 0.06 (−0.02 to 0.14) | 0.162 |
| Junior high school | 0.13 (0.04 to 0.22) | 0.005 | 0.0001 (−0.10 to 0.10) | 0.998 |
| High school or above | 0.22 (0.11 to 0.33) | <0.001 | 0.09 (−0.03 to 0.20) | 0.149 |
| Presenting VA in better-seeing eye | | | | |
| ≥6/12 | Reference | | Reference | |
| ≥6/18 to <6/12 | −0.04 (−0.14 to 0.07) | 0.496 | −0.005 (−0.10 to 0.09) | 0.911 |
| ≥6/60 to <6/18 | −0.08 (−0.18 to 0.01) | 0.080 | −0.02 (−0.11 to 0.07) | 0.707 |
| <6/60 | 0.05 (−0.21 to 0.30) | 0.718 | 0.17 (−0.08 to 0.42) | 0.173 |
| Standard residential health insurance | −0.15 (−0.21 to to 0.09) | <0.001 | −0.08 (−0.15 to 0.004) | 0.040 |
| Self-assessment of health | | | | |
| Bad | Reference | | Reference | |
| Not good | −0.27 (−0.47 to to 0.07) | 0.009 | −0.13 (−0.47 to 0.21) | 0.455 |
| Good | −0.22 (−0.42 to 0.02) | 0.028 | −0.07 (−0.43 to 0.28) | 0.696 |
| Very good | −0.26 (−0.46 to 0.06) | 0.010 | −0.08 (−0.43 to 0.28) | 0.672 |
| Excellent | −0.22 (−0.47 to 0.02) | 0.073 | −0.32 (−0.73 to 0.10) | 0.134 |
| Self-assessment of VA | | | | |
| Bad | Reference | | Reference | |
| Not good | −0.28 (−0.48 to to 0.08) | 0.007 | −0.11 (−0.46 to 0.24) | 0.532 |
| Good | −0.23 (−0.43 to 0.04) | 0.018 | −0.05 (−0.41 to 0.31) | 0.781 |
| Very good | −0.27 (−0.47 to 0.07) | 0.008 | −0.06 (−0.42 to 0.31) | 0.756 |
| Excellent | −0.03 (−0.29 to 0.23) | 0.835 | 0.33 (−0.10 to 0.76) | 0.135 |
| Economic situation | | | | |
| <30% income paid for food | Reference | | | |
| ≥30% income paid for food | 0.007 (−0.07 to 0.08) | 0.847 | | |

*The amount willing to pay was not normally distributed and logarithmic transformation was applied.
†Age, sex and variables with p<0.20 in the univariable regression analysis were included in the multivariable regression model.
VA, visual acuity; β, parameter coefficient estimate.

paid for by the government; cataract surgeries are in most cases associated with significant improvements in vision; and coverage rates of medical insurance have increased in China since 2008, potentially reducing WTP out of pocket for healthcare services.

Many countries have experimented with DR screening models to reduce barriers to access and increase service uptake. For example, the UK and some other European countries have established free community screening for DR[21 22] and found it to be cost-effective.[23] Teleophthalmology and general practitioners have been used in the USA, with services either reimbursed by insurance, or paid fully or in part out-of-pocket.[24] General practitioners' referrals for those detected at DR screenings to retinal specialists for further treatment were shown to be effective in preventing VI from DR in Australia.[25 26]

There are also established DR screening services in Asian countries such as Singapore,[27] India[27 28] and Taiwan.[29]

This study suggests that rural residents and those who are less economically advantaged are more willing to pay for DRS services. In Qujiang District, Shaoguan City, rural residents with DM would otherwise have to travel to the neighbouring district hospital or the city centre to seek eye services. Given that there is no public transportation, this trip would take over an hour. Travelling by motorcycle was the most common mode used by rural residents, making an hour-long trip difficult. In addition, many elderly could only be taken to the city or the neighbouring district with the help of a younger person. The fact that this programme increased ease of access by bringing DRS services into rural areas may explain why rural residents were more likely to be willing to pay for screening at their township health unit.

Our study found that people covered by employment-linked insurance were more willing to pay higher fees for DRS. This is likely due to their having comparatively better medical insurance and possibly higher incomes. Unfortunately, it was not possible to collect information on income to test this hypothesis in the current study.

In the current study, men and urban residents were willing to pay more for DRS. This could potentially be explained by their being more economically advantaged on average, or their having greater awareness of the importance of DRS. Urban residents may be more apt to understand the need for examinations to prevent diabetic complications and this suggests that future health education interventions should target rural residents, especially rural women.

Another study has reported that more severe DR among persons with type 2 diabetes was associated with lower perceived utility from the patient's preference-based viewpoint.[30] We did not analyse this metric as the number of patients at R3 was not great enough (9 people, 1.7% among the 545 participants). Rather, we compared the self-assessment of VA and WTP, but found no association between WTP and self-assessed health or vision.

Strengths of the current study include its relevance to actual screening activities, having been community based and incorporated into a scheduled, large-scale DR screening programme supported by the local hospital, health authorities and primary level health staff. In addition, to determine the sum participants were willing to pay, we used a direct method with payment cards of differing amounts which were presented simultaneously to participants for selection. This avoided starting point bias in bidding for payment. To avoid range bias,[31] we used ¥10 (US$1.6) for a small-scale stable range which would not greatly affect the payment value. The minimum value of ¥10 is equal to the common consultant fee for a hospital outpatient department, and the maximum value of ¥120 (US$18.8) is equal to the sum charged for two-eye fundus images taken at secondary-level hospitals. The presented value on payment cards was empirical and conceptual, taking account of the actual pricing of ocular medical services as well as the inherent monetary value of ¥10 (US$1.6).

Some weaknesses must also be acknowledged. The study was conducted at a single point in time and cannot reflect long-term WTP values as the economy is constantly developing. The study also took place in one mostly rural region, so it does not represent other areas in China. This WTP study is hypothetical and assumes that screening is beneficial to the health of participants. Cost of screening was introduced and reinforced by eliciting payment amounts. We have also assumed that the respondents were willing to accept the screening as a benefit to their health. The amounts which participants reported they were willing to pay and the amounts they would actually pay may not be the same.[31]

To our knowledge, this is the first study on WTP for DRS in China. Cost has been highlighted as the major barrier to establishing DRS at the primary level in China, outside of wealthy areas such as Shanghai, where DRS is well established and fully covered by the metropolitan city government. Findings from this WTP study provide evidence that substantial screening costs could potentially be covered for by patients.

While a considerable percentage of interviewed PwDM demonstrated WTP on some personal cost for DRS, studies have noted that organised screening programmes result in substantial cost savings for national health systems.[19 30] Recognising that some 50% of current national medical expenditures in China are covered by government medical insurance, investment to establish systematic DRS at the primary care level could substantially reduce costs for the national healthcare system, as well as reducing out-of-pocket medical expenditures.[19] Conducting additional studies in urban areas, as well as in other diverse settings, would contribute to a more complete picture of WTP for DRS in China and help guide the establishment of organised DRS programmes. To all the stakeholders of this project, sustainability of the current programme would take a key role to serve PwDM in the areas.

## CONCLUSIONS

Nearly two-thirds of participants were willing to pay for screening in this screening programme event organised at the care primary level in rural China, which offers the potential that such activities can be sustained and scaled up through user fees.

**Acknowledgements** The authors acknowledge the doctors and nurses from all the ten township health units involved to liaise with the local patients, screening site coordination. Financial support from Shaoguan Rural Health Express program was crucial to the success of the DRS in Qujiang. The trained screening team including Ms Qiong Wan as the team leader from Shaoguan Railway Hospital (Zhenjiang District Hospital) was greatly appreciated.

**Contributors** BX: Designed and monitored the whole study, initially drafted and revised the manuscript. LJ: Analyaed data. XG, VFC, YL and DL: Critically commented and revised the manuscript. CP-S: Corrected English and critically commented on the manuscript. YL, YW, HF: Coordinated data collection and commented critically on the manuscript. NC: Guided the study generally and revised the manuscript. BX is the author as the guarantor.

**Funding** The study was part of 'Diabetes Phase II project' funded by Orbis International (reference no. NA), World Diabetes Foundation (WDF15-1290), Shaoguan Rural Health Express Project (reference no. NA) and the Zhenjiang District Hospital (reference no. NA). NC received Wellcome Trust funding.

**Competing interests** None declared.

**Patient and public involvement** Patients and/or the public were involved in the design, or conduct, or reporting, or dissemination plans of this research. Refer to the Methods section for further details.

**Patient consent for publication** Consent obtained directly from patient(s).

**Ethics approval** Ethical approval for this study was obtained from Zhongshan Ophthalmic Center, Sun Yat-sen University. (2019KYPJ067). Participants gave informed consent to participate in the study before taking part.

**Provenance and peer review** Not commissioned; externally peer reviewed.

**Data availability statement** Data are available on reasonable request from the first and corresponding authors.

**ORCID iDs**
Baixiang Xiao http://orcid.org/0000-0003-1987-4851
Ving Fai Chan http://orcid.org/0000-0002-4968-7953

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
