## [Reviewer comments · BMJ Open]

ARTICLE DETAILS

TITLE (PROVISIONAL)	Willingness to pay for diabetic retinopathy screening in Qujiang District, rural Guangdong, southern China: a cross-sectional study
AUTHORS	Congdon, Nathan; Xiao, Baixiang; Gu, Xuejun; Jin, Ling; Chan, Ving; Li, Yanping; Price-Sanchez, Carlos; Liu, Yuanping; Wang, Yanfang; Fu, Haoxiang; Li, Dongfeng

VERSION 1 – REVIEW

REVIEWER	Bayable, Samuel Debas Debre Markos University
REVIEW RETURNED	15-Dec-2022

GENERAL COMMENTS	Generally, I would like to appreciate your contribution to the scientific world and the paper is well written and informative with the following comments. General comments Revise the journal policies and editors' process from submission to publicationRevise typos errors, grammatical problems, and plagiarism throughout the whole document.Avoid unnecessary details throughout the whole document (methods section).You are using a lot of outdated references, so please try to minimize them. Major comments o The abstract methods section is disorganized and unclear if someone reads only your abstract, so please rearrange and add missed crucial points, like analysis.....o In the result section, it is not good to elaborate on every finding of the tables section in the text.o Economic status was statically significant but the subjective classification of economic status may end up your finding to the wrong conclusion, how can I confirm it is free from the researcher bias?o Even if a lot of objective assessments were conducted during retinopathy screening, you are concentrating more on subjective assessment. Is there any possible justification? Minor comments o The trained data collector was a single person, how you are going to achieve such huge data with one data collector?o Since this research is part of the project what are your recommendations for the project-leading organization or the community at large?
--

REVIEWER	Gakunga, Robai Independent Research Scientist
REVIEW RETURNED	16-Dec-2022

GENERAL COMMENTS	This is a great paper. The research question is clear, the methodology is presented very well and analysis is well done. A few clarifications may make it better; -Results sections (in the abstract and the body): You have mentioned that men, urban residents, and those covered by employer linked insurance were willing to pay higher amounts than their counterparts. How about those with high schooling and above, as well as those with bad health and VA? Should they be included among those willing to pay higher amounts than their counterparts? It would be great to include the reasons given by the 40% who answered "no" to paying something. Please revise lines 6-12 (Page 12) Did you mean: linear regression analysis of participant characteristics in association with the level of willingness to pay?
---

REVIEWER	Emire , Mamo Solomon Wolkite University College of Medicine and Health Science
REVIEW RETURNED	20-Dec-2022

GENERAL COMMENTS	Authors must fulfil the following points  [ ] The abstract conclusion lacks important aspects in the results [ ] Consent Authors do not clearly take written informed consent from individual participants. [ ] Reference: The majority of references are out of date. [ ] The discussion section of the paper does not clearly describe and compare with other similar studies, nor does it explain why the results were obtained. [ ] The conclusion is crucial need rewrite again. recommend for this manuscript, which will be extensively revised before publication
---

REVIEWER	Kelly, Stephen R. Mater Misericordiae University Hospital, Mater Retina Research Group
REVIEW RETURNED	30-Jan-2023

GENERAL COMMENTS	A well written manuscript of an important study area. Some comments: Abstract: mostly fine, first sentence in "Design" could be rephrased. Similarly second sentence, replace "from among". Strengths and limitations: rephrase first sentence "incorporated to actual scheduled.." Introduction: DM is defined but then "Diabetes mellitus" is still used on line 25. Similarly, diabetes and DM are used interchangeably, might be best to stick with one. Maybe rephrase "willing to pay *anything*" with willing to pay something p18 L12 and throughout. Table 3 says willing to pay "something" which seems a better approach. Pay some amount vs pay anything.
--

	The results find that those with DR complications are less likely to pay (although those that do pay, are willing to pay slightly more). It would be useful to see the breakdown from different complications if possible. Would people with DR specific complications be willing to pay? It would be valuable to include an explanation of this result from the authors. I guess I'm unclear on the rural vs urban divide, there were 10 centres with some being rural and urban. As the authors mention, travel time/cost can be an issue for people. Were all of these appointments within walking distance? Did the people in the study have to pay the amount they stated they would be willing to pay? I assume not but would be good to clarify this. The introduction states the data collected included profession and comorbidities. It might be useful to include these in the model in some way. Perhaps "labour/farming/agricultural" and other. Certain jobs may have a larger cost of attending due to taking time off/losing pay. People with other chronic conditions, who have more regular contact with health care, may be more/less willing to pay. The study also lists venous blood samples were collected. Did blood glucose/HbA1c levels affect the willingness to pay? Might also be worth adding to the model.
--	--

VERSION 1 – AUTHOR RESPONSE

Reviewer: 1

Dr. Samuel Debas Bayable, Debre Markos University

Comments to the Author:

Generally, I would like to appreciate your contribution to the scientific world and the paper is well written and informative with the following comments.

Response: Thanks.

General comments

Revise the journal policies and editors' process from submission to publication

Revise typos errors, grammatical problems, and plagiarism throughout the whole document.

Response: Thanks. This has been done.

Avoid unnecessary details throughout the whole document (methods section).

You are using a lot of outdated references, so please try to minimize them.

Response: The MS has been revised to address these points.

Major comments

- The abstract methods section is disorganized and unclear if someone reads only your abstract, so please rearrange and add missed crucial points, like analysis.....

Response: This has been revised.

• In the result section, it is not good to elaborate on every finding of the tables section in the text.

Response: Thanks for the good point. The Results section has now been focused.

o Economic status was statically significant but the subjective classification of economic status may end up your finding to the wrong conclusion, how can I confirm it is free from the researcher bias?

Response: We have reduced the number of economic categories from 5 to 2, choosing the most natural dividing point in the distribution.

o Even if a lot of objective assessments were conducted during retinopathy screening, you are concentrating more on subjective assessment. Is there any possible justification?

Response: The authors are not clear that we understand this comment. This study was based on a program in which objective assessments of the eye were carried out, but the purpose of this paper is to characterize willingness to pay and its determinants. This was done in an objective, carefully described fashion in order to reduce bias to the fullest possible extent.

Minor comments

o The trained data collector was a single person, how you are going to achieve such huge data with one data collector?

Response: For the general information in the study, we had 3 interviewers, for willingness to pay, there was only one.

o Since this research is part of the project what are your recommendations for the project-leading organization or the community at large?

Response: These are now added

Reviewer: 2

Dr. Robai Gakunga, Independent Research Scientist

Comments to the Author:

This is a great paper.

The research question is clear, the methodology is presented very well and analysis is well done.

Response: Thanks.

A few clarifications may make it better;

-Results sections (in the abstract and the body):

You have mentioned that men, urban residents, and those covered by employer linked insurance were willing to pay higher amounts than their counterparts. How about those with high schooling and above, as well as those with bad health and VA? Should they be included among those willing to pay higher amounts than their counterparts?

Response: Thanks for these questions. These other factors were tested and no significant differences were detected in our analyses..

It would be great to include the reasons given by the 40% who answered "no" to paying something.

Response: These are now added.

Please revise lines 6-12 (Page 12)

Response: Done

Did you mean: linear regression analysis of participant characteristics in association with the level of willingness to pay?

Response: Correct. This has been clarified.

Reviewer: 3

Dr. Mamo Solomon Emire , Wolkite University College of Medicine and Health Science

Comments to the Author:

Authors must fulfil the following points

The abstract conclusion lacks important aspects in the results

Response: The authors strongly disagree. The fact that a sufficient number of patients were willing to pay for eye screening suggests that this activity can be scaled up sustainably, which we feel is the most important conclusion of this study.

Consent Authors do not clearly take written informed consent from individual participants.

Response: In fact, the Methods states: "Written consent was obtained from each participant."

Reference: The majority of references are out of date.

Response: The authors feel that references represent the current literature, but would be happy to include any specific new references the reviewer would like to suggest.

The discussion section of the paper does not clearly describe and compare with other similar studies, nor does it explain why the results were obtained.

Response: The Discussion section in fact does compare our results with the limited number of other similar papers, and describes how we searched to find these. We have offered our suggestions about a number of factors, including most of our findings with respect to the drivers of WTP, and why WTP for DR screening appears lower than for cataract surgery.

Reviewer: 4

Dr. Stephen R. Kelly, Mater Misericordiae University Hospital

Comments to the Author:

A well written manuscript of an important study area.

Some comments:

Abstract: mostly fine, first sentence in "Design" could be rephrased. Similarly second sentence, replace "from among".

Response: Done

Strengths and limitations: rephrase first sentence "incorporated to actual scheduled.."

Response: Revised

Introduction: DM is defined but then “Diabetes mellitus” is still used on line 25. Similarly, diabetes and DM are used interchangeably, might be best to stick with one.

Response: Revised. Thanks

Maybe rephrase “willing to pay *anything*” with willing to pay something p18 L12 and throughout. Table 3 says willing to pay “something” which seems a better approach. Pay some amount vs pay anything.

Response: We have tried to be consistent in using “willing to pay something,” but in the negative, we have used “not willing to pay anything.”

The results find that those with DR complications are less likely to pay (although those that do pay, are willing to pay slightly more). It would be useful to see the breakdown from different complications if possible. Would people with DR specific complications be willing to pay? It would be valuable to include an explanation of this result from the authors.

Response: Thanks. It is now added.

I guess I'm unclear on the rural vs urban divide, there were 10 centres with some being rural and urban. As the authors mention, travel time/cost can be an issue for people. Were all of these appointments within walking distance?

Response: This varied between participants, but it is clear that the screening locations were considerably more convenient than current alternatives, especially for rural patients.

Did the people in the study have to pay the amount they stated they would be willing to pay? I assume not but would be good to clarify this.

Response: Please note that the following was present in the original version of the Methods: “Each participant was informed that their choice of willingness to pay category would not affect their ability to receive primary health care, medical services in hospitals, nor current or any future DR screening”.

The introduction states the data collected included profession and comorbidities.

It might be useful to include these in the model in some way. Perhaps “labour/farming/agricultural” and other. Certain jobs may have a larger cost of attending due to taking time off/losing pay. People with other chronic conditions, who have more regular contact with health care, may be more/less willing to pay.

Response: yes, those with more severe situation were rather going to hospital. These are now added to the reasons of unwilling to pay.

The study also lists venous blood samples were collected. Did blood glucose/HbA1c levels affect the willingness to pay? Might also be worth adding to the model.

Response: Yes, we did, but did not discover a significant association, and so did not present the data while results added now.

VERSION 2 – REVIEW

REVIEWER	Emire , Mamo Solomon Wolkite University College of Medicine and Health Science
REVIEW RETURNED	08-Mar-2023

GENERAL COMMENTS	conclusion in the abstract needs detail description? How data collection took place five years ago, to just see if anything related has been published with this title. PwDM increased from about 2% in the 1980s to over 10% in 2017, attempting to make the years 2022 and 2023 a better compare for the current. It is preferable to discuss the severity of diabetic retinopathy than the number of people worldwide who have the DM in China. Your study in rural areas, how significance urban residents were willing to pay more for DRS. Strengths of the current study include its relevance to actual screening activities, having been community-base this is your routine activity of researcher but not strength of this study. This study focus is placed more on cost than the course of the disease from beginning to end.so better to study on DRS You write a conclusion and its relevance in the abstract, but not in the main manuscript. Some references still require updating because they are out-dated
--

REVIEWER	Kelly, Stephen R. Mater Misericordiae University Hospital, Mater Retina Research Group
REVIEW RETURNED	10-Mar-2023

GENERAL COMMENTS	Most of the concerns have been addressed but others (such as asking for a breakdown of DR specific complications or suggested inclusions to the model) have not. The authors responded saying the changes were implemented but I don't see any in the revised manuscript.
---

VERSION 2 – AUTHOR RESPONSE

Reviewer: 3

Dr. Mamo Solomon Emire , Wolkite University College of Medicine and Health Science

Comments to the Author:

Conclusion in the abstract needs detail description?

How data collection took place five years ago, to just see if anything related has been published with this title.

Response: Thanks for the reminding. We surveyed online and did not detected new publication with this title.

PwDM increased from about 2% in the 1980s to over 10% in 2017, attempting to make the years 2022 and 2023 a better compare for the current.

Response: The huge study was conducted in 2017, while there is no updated ones owing to the COVID-19.

It is preferable to discuss the severity of diabetic retinopathy than the number of people worldwide who have the DM in China.

Your study in rural areas, how significance urban residents were willing to pay more for DRS.

Response: There are also towns in this area.

Strengths of the current study include its relevance to actual screening activities, having been community-base this is your routine activity of researcher but not strength of this study.

Response. This is understood and incorporated into the large screening gave more credit so it is still the advantage.

This study focus is placed more on cost than the course of the disease from beginning to end.so better to study on DRS

You write a conclusion and its relevance in the abstract, but not in the main manuscript.

Response: Thanks it is the same as in the manuscript.

Some references still require updating because they are out- dated

Response: The evidence gained from developed countries for DRS was good references to the situation in China and the fact not changed, so we still use them as the references.

Reviewer: 4

Dr. Stephen R. Kelly, Mater Misericordiae University Hospital

Comments to the Author:

Most of the concerns have been addressed but others (such as asking for a breakdown of DR specific complications or suggested inclusions to the model) have not. The authors responded saying the changes were implemented but I don't see any in the revised manuscript.

Below is the comments from reviewer 4 at the first time, and the response

Reviewer: 4

Dr. Stephen R. Kelly, Mater Misericordiae University Hospital

Comments to the Author:

A well written manuscript of an important study area.

Some comments:

Abstract: mostly fine, first sentence in "Design" could be rephrased. Similarly second sentence, replace "from among".

Response: Done

Strengths and limitations: rephrase first sentence "incorporated to actual scheduled.."

Response: Revised

Introduction: DM is defined but then "Diabetes mellitus" is still used on line 25. Similarly, diabetes and DM are used interchangeably, might be best to stick with one.

Response: Revised. Thanks

Maybe rephrase "willing to pay *anything*" with willing to pay something p18 L12 and throughout. Table 3 says willing to pay "something" which seems a better approach. Pay some amount vs pay anything.

Response: We have tried to be consistent in using “willing to pay something,” but in the negative, we have used “not willing to pay anything.”

The results find that those with DR complications are less likely to pay (although those that do pay, are willing to pay slightly more). It would be useful to see the breakdown from different complications if possible. Would people with DR specific complications be willing to pay? It would be valuable to include an explanation of this result from the authors.

Response: Thanks. It is now added on page 16 under table 4 and above discussion session.

I guess I'm unclear on the rural vs urban divide, there were 10 centres with some being rural and urban. As the authors mention, travel time/cost can be an issue for people. Were all of these appointments within walking distance?

Response: This varied between participants, but it is clear that the screening locations were considerably more convenient than current alternatives, especially for rural patients.

Did the people in the study have to pay the amount they stated they would be willing to pay? I assume not but would be good to clarify this.

Response: Please note that the following was present in the original version of the Methods: “Each participant was informed that their choice of willingness to pay category would not affect their ability to receive primary health care, medical services in hospitals, nor current or any future DR screening”.

The introduction states the data collected included profession and comorbidities.

It might be useful to include these in the model in some way. Perhaps "labour/farming/agricultural" and other. Certain jobs may have a larger cost of attending due to taking time off/losing pay. People with other chronic conditions, who have more regular contact with health care, may be more/less willing to pay.

Response: yes, those with more severe situation were rather going to hospital. These are now added to the reasons of unwilling to pay on page 16 above discussion session.

The study also lists venous blood samples were collected. Did blood glucose/HbA1c levels affect the willingness to pay? Might also be worth adding to the model.

Response: Yes, we did put HaA1c into the model, but did not discover a significant association, and so did not present the data while results added now.

VERSION 3 – REVIEW

REVIEWER	Kelly, Stephen R. Mater Misericordiae University Hospital, Mater Retina Research Group
REVIEW RETURNED	31-Mar-2023
GENERAL COMMENTS	Happy to accept if CROSS checklist is complete.